

## Effect of NOx on 1,3,5-trimethylbenzene (TMB) oxidation product distribution and particle formation

Julia Hammes[1], Epameinondas Tsiligiannis[1], Thomas F. Mentel[1,2], Mattias Hallquist[1]*

[1]Department of Chemistry and Molecular Biology, University of Gothenburg, Gothenburg, Sweden

[2]Institute of Energy and Climate Research, IEK-8: Troposphere, Forschungszentrum Jülich GmbH, Jülich, Germany

* *Correspondence to*: hallq@chem.gu.se

## Abstract

Secondary organic aerosol (SOA) represents a significant fraction of the tropospheric aerosol and its precursors are volatile organic compounds (VOC). Anthropogenic VOCs (AVOC) dominate the VOC budget in many urban areas with 1,3,5-trimethylbenzene (TMB) being among the most reactive aromatic AVOCs. TMB formed highly oxygenated organic molecules (HOM) in $NO_x$ free environment, which could contribute to new particle formation (NPF) depending on oxidation conditions were elevated OH oxidation enhanced particle formation. The experiments were performed in an oxidation flow reactor, the Go:PAM unit, under controlled OH oxidation conditions. By addition of $NO_x$ to the system we investigated the effect of $NO_x$ on particle formation and on the product distribution. We show that the formation of HOM and especially HOM accretion products, strongly varies with $NO_x$ conditions. We observe a suppression of HOM and particle formation with increasing $NO_x/\Delta TMB$ and an increase in the formation of organonitrates (ON) mostly at the expense of HOM accretion products. We propose reaction mechanisms/pathways that explain the formation and observed product distributions with respect to oxidation conditions. We hypothesize that, based on our findings from TMB oxidation studies, aromatic AVOCs may not contribute significantly to NPF under typical $NO_x$/AVOC conditions found in urban atmospheres.

## 1    Introduction

Volatile organic compounds (VOC) are ubiquitous in the atmosphere and major precursors for secondary organic aerosol (SOA). SOA represents a dominant fraction of the tropospheric aerosol (Hallquist et al., 2009;Shrivastava et al., 2017;Gentner et al., 2017) and affects climate (Intergovernmental Panel on Climate, 2014) and health (WHO, 2016). Consequently research interest in SOA formation and properties is ranging from remote atmospheres (Ehn et al., 2012;Ehn et al., 2014;Kristensen et al., 2016) to densely populated and polluted environments (Chan and Yao, 2008;Hu et al., 2015;Guo et al., 2014;Hallquist et al., 2016). Following the study by Ehn et al. (Ehn et al., 2012), highly oxygenated organic molecules (HOM) with low volatilities, formed from the oxidation of biogenic volatile organic compounds (BVOCs), have attracted much research interest (Crounse et al., 2013;Ehn et al., 2014;Jokinen et al., 2014;Jokinen et al., 2015;Mentel et al., 2015;Yan et al., 2016;Berndt et al., 2016;Bianchi et al., 2019). These compounds have been shown to contribute to new particle formation (NPF) and to SOA growth (Ehn et al., 2014;Bianchi et al., 2016;Kirkby et al., 2016;Trostl et al., 2016;McFiggans et al., 2019),



making them an important factor in the formation of atmospheric SOA. These oxidation products can be either described as HOM based on their high oxygen number (O> 6) (Bianchi et al., 2019) or as extremely low volatile organic compounds (ELVOC) based on their volatility (Donahue et al., 2012;Trostl et al., 2016). In this study, we will refer to oxidation products as HOM, because not all of the measured compounds may fulfill the criteria for

ELVOC (Trostl et al., 2016;Kurtén et al., 2016). Gas phase autoxidation of alkylperoxy radicals ($RO_2$) has been proposed as the formation mechanism for HOM (Crounse et al., 2013;Ehn et al., 2014;Jokinen et al., 2014). After the initial reaction of an oxidant with the VOC and subsequent addition of $O_2$ to the alkylradical (R), the produced $RO_2$ isomerizes via intra molecular H abstraction (H-shift). During this process a hydroperoxide group and a new R is formed. Additional $O_2$ addition and H-shift sequences can introduce large amounts of oxygen to the molecule

and subsequently lower the vapour pressure. The chemistry of aromatic compounds is somewhat different compared to other VOCs as they can lose their aromaticity during the initial OH attack while they can retain the ring structure. Moreover, reaction products are more reactive than the parent compound. The produced $RO_2$ form an oxygen bridge, a bicyclic and potentially a tricyclic alkylradical (Molteni et al., 2018;Wang et al., 2017) before further oxidation processes open the ring structure. For 1,3,5-trimethylbenzene (TMB), emitted from combustion

sources in the urban environment, Molteni et al. (2018) proposed a generalized reaction scheme for HOM formation after OH addition. According to their scheme, first generation alkylperoxy radicals with the general formula of $C_9H_{13}O_{5-11}$ were formed from the initial OH attack and subsequent H shift and $O_2$ addition sequences. A postulated second OH attack would result in propagating peroxy radical chains yielding radicals with the general formulas $C_9H_{15}O_{7-11}$.

Generally, the termination reaction of $RO_2$ (with a general $m/z$ = x) with $HO_2$, leads to the formation of hydro peroxides ($m/z$ = x +1) while termination reactions with other $RO_2$ can lead to the formation of a carbonyl ($m/z$ = x -17), a hydroxy group ($m/z$ = x -15) or dimers ($m/z$ = 2x – 32) (Mentel et al., 2015;Jokinen et al., 2014;Rissanen et al., 2014). The propagation reaction of $RO_2$ with another $RO_2$ or NO also results in the formation of RO ($m/z$ = x-16). The RO can undergo internal H shift leading to the formation of an hydroxy group and

subsequent a new peroxy radical which can continue the autoxidation sequences. The alkoxy step shifts of the observed m/z by 16 leads to overlap of different termination product sequences. During extensive oxidation the first generation products are subject to secondary chemistry increasing the numbers of products. Molteni et al. (Molteni et al., 2018) found several closed shell monomer products with the general formula $C_9H_{12-16}O_{5-11}$ from the oxidation of TMB with OH. The formation of dimers with different number of H atoms in their study was explained

by reactions of two first generation $RO_2$ radicals ($C_{18}H_{26}O_{5-10}$), one first and one second generation $RO_2$ ($C_{18}H_{28}O_{9-12}$) or two second generation $RO_2$ resulting in the dimer $C_{18}H_{30}O_{11}$. Although dimers have in general lower O:C ratios than monomers, they are expected to be less volatile due to higher molecular weight and more functional groups making them candidates to participate in nucleation processes (Kirkby et al., 2016).

Organonitrates (ON) are formed as soon as sufficient $NO_x$ is present in the atmosphere. ON are highly

important for the reactive nitrogen budget wherein the formation of highly functionalized organic nitrates can contribute significantly to secondary organic aerosol (Lee et al., 2016;Bianchi et al., 2017). In this study we refer to compounds that only consist of H, C and O as HOM monomers or HOM dimers and to N containing compounds as ONs.



NO$_x$ influences the oxidation of organics directly by changing oxidant levels (reducing or increasing OH, depending on the NO$_x$ regime) and indirectly by influencing RO$_2$ chemistry. In high NO$_x$ environment such as urban areas the reaction of NO with RO$_2$ radicals

$$RO_2 + NO \rightarrow RONO2 \tag{1}$$
$$\rightarrow RO + NO_2 \tag{2}$$

can compete with the autoxidation mechanism (reaction 1 and 2). and thus potentially inhibit HOM while favouring ON formation (reaction 1). Although ON may have high O:C they differ from HOM as they contain at least one nitrogen atom. While HOM formation from BVOCs has been intensively studied, only few studies have been conducted focusing on the formation of HOM from anthropogenic aromatic volatile organic compounds (AVOCs) (Molteni et al., 2018;Wang et al., 2017). However, these studies indicate that AVOCs have a strong potential to form HOM under NO$_x$ free conditions and proposed that they may play a crucial role in NPF and particle growth of SOA in urban areas. Regarding SOA yields a number of smog chamber studies have been conducted in order to investigate the oxidation of TMB with OH radical under different NO$_x$ and aerosol seed conditions (Paulsen et al., 2005;Rodigast et al., 2017;Wyche et al., 2009;Sato et al., 2018;Huang et al., 2015). They reported that under higher NO$_x$:VOC conditions, SOA yield is reduced compared to medium or lower NO$_x$:VOC conditions. The NO$_x$:VOC, in almost all studies, was < 1, apart from one experiment (Wyche et al., 2009), with NO$_x$:VOC = 1.9 in which the SOA yield was 0.29, compared to yields up to 7.47 under lower NOx conditions.

In this study we investigate the oxidation of TMB in a laminar flow reactor, while different NO$_x$ and OH conditions were applied. A nitrate chemical ionization atmospheric pressure interface time of flight mass spectrometer (CI-APi-TOF-MS) (Junninen et al., 2010;Jokinen et al., 2012) was used to monitor the oxidation product distribution. We show the formation of HOM and nitrate containing compounds with and without NOx added to the reaction system. Possible mechanisms leading to the formation of ON and suppression of particle formation are discussed.

## 2 Materials and Methods

The measured HOM were generated using the laminar flow Gothenburg Potential Aerosol Mass reactor (Go:PAM), initially described by Watne et al. (Watne et al., 2018). The Go:PAM is a 100 cm long, 9.6 cm wide quartz glass cylinder which is irradiated over a length of 84 cm by two 30 W Phillips TUV lamps (254 nm); a schematic is shown in Figure 1. The OH radicals are produced inside Go:PAM by photolysing O$_3$ in the presence of water vapour. The O$_3$ is generated outside Go:PAM by photolyzing pure O$_2$ (UVP Pen-Ray® Mercury Lamps, 185 nm) and distributed in 3 L min$^{-1}$ particle free and humidified air (Milli – Q) over the reactor cross section. The VOC was introduced through a gravimetrically characterized diffusion source (see SI Figure 1) centrally at the top of the reactor with a flow of 8 L min−1. Flows were adjusted for a median residence time of 34 s in Go:PAM. A funnel shaped device is subsampling the centre part of the laminar flow to minimize wall effects on the sample. A condensation particle counter (CPC, 3775 TSI) was used to measure the number particle concentration in the sample flow. O$_3$ was monitored by a model 202 monitor (2B Technologies), relative humidity by a Vaisala HMP60 probe and NO$_x$ by a model 42i monitor (Thermo Scientific) over the course of the experiments. The OH exposure, over the residence time in the reactor, for NO$_x$ free conditions without added TMB was measured using SO$_2$ titration



(Teledyne T100 ) as described by Kang et al. (Kang et al., 2007). Oxidation products were measured with an Atmospheric Pressure interface High Resolution Time of Flight Mass Spectrometer (APi-TOF-MS, Aerodyne Research Inc. & Tofwerk AG) (Junninen et al., 2010;Jokinen et al., 2012) in connection with a A70 CI-inlet (Airmodus Ltd) (Eisele and Tanner, 1993). The CI inlet is a laminar flow inlet operated with a sheath flow of 20 L

5    min$^{-1}$ containing $NO_3^-$ ions which are generated by ionizing $HNO_3$ using an $^{241}$Am foil upstream in the inlet design. The sample stream from Go:PAM is introduced in the centre of the sheath flow at a rate of 8 L min$^{-1}$. The $NO_3^-$ ions are electrostatically pushed into the sample flow and form stable adducts with sample molecules as described by Ehn et al. (2012). The reaction time of oxidation products and $NO_3^-$ is a few hundred ms before being subsampled into the TOF-MS at 0.8 L min$^{-1}$ by a critical orifice. Differential pumping decreased the pressure from 103 mbar

10    in the CI source to $10^{-6}$ mbar in the TOF extraction region where HOM are detected as negatively charged clusters with $NO_3^-$.

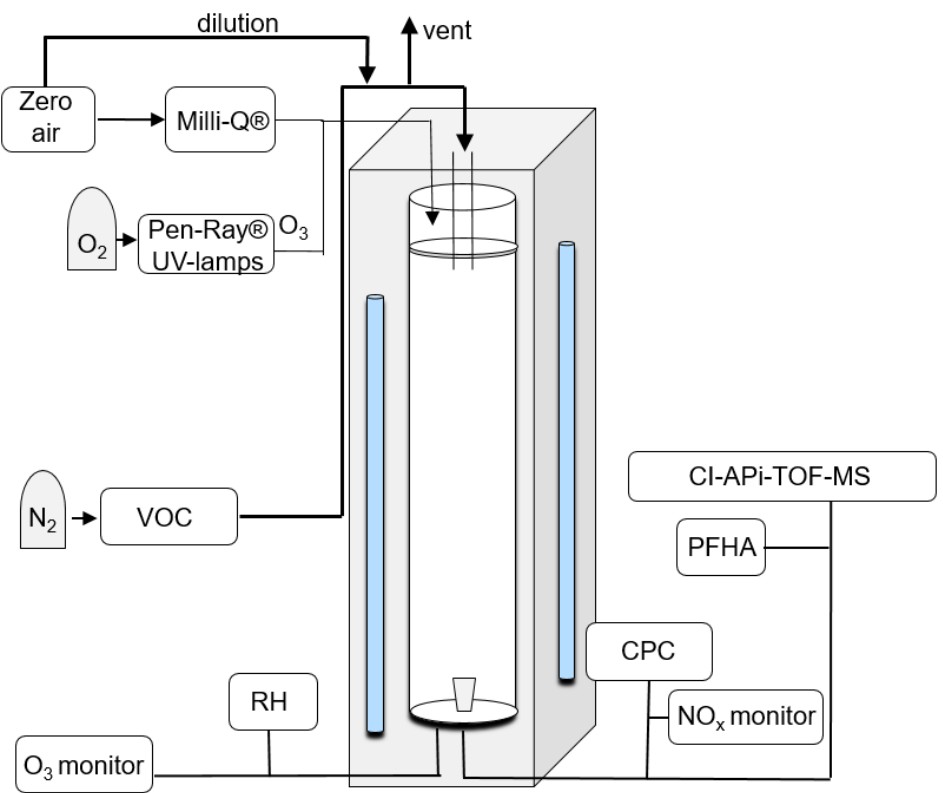

**Figure 1: Schematics of the experimental setup with Go:PAM chamber connected to CI- APi-TOF-MS.**

     A kinetic box model was used to simulate the chemistry in the Go:PAM reactor. The core of the model

15    were first described by Watne et al. (Watne et al., 2018). The model consists of 32 species and 68 reactions now including TMB chemistry partly from the MCM v3.3.1 (Jenkin et al., 2003) as well as proposed mechanisms and rate coefficients for $NO_2$ chemistry (Atkinson et al., 1992;Finlayson-Pitts, 1999) and highly oxygenated compounds (Ehn et al., 2014; Berndt et al., 2018; Zhao et al., 2018) (see SI 1). The photon flux used in the simulations was tuned to match measured decay of $O_3$ while an OH sink was added to match the observed OH exposure in the





background experiment, i.e. without the addition of TMB. The model was run for all experiments with and without $NO_x$. Primarily, the modelled output on OH exposure for each experiments was used to interpret the results and for calculating the consumed TMB. However, broadly the modelling output was also used to understand the effects of changing experimental conditions on monomer, dimer and organo nitrate (ON) production. The experiments

without $NO_x$ were named 1-4 denoting the increase in OH exposure and experiments with $NO_x$ were denoted according to their $NO_x$/ $\Delta$TMB and high (H) and low (L) OH exposure as seen in Table 1.

### 3    Results and Discussions

Table 1 summarises eight experiments where TMB has been oxidised by various amounts of OH using the Go:PAM unit. Generally, a high OH production, induced either by increased light exposure (two lamps) or elevated ozone

concentration, resulted in new particles (e.g. exp 3 and 4) while addition of $NO_x$ reduced or suppressed the particle formation. The results from the kinetic model show that the amount of reacted TMB ranges from 5 – 30 ppb, depending on OH exposure (SI Figure 1). An overview of the oxidation product distribution measured with the CI-APi-TOF-MS for different conditions is shown in Figure 3 and Figure 4. The compounds were detected as nitrate clusters at $m/z$ = mass$_{compound}$ + 62. The spectra in Figure 2 and 3 show significant ion signals from oxygenated

hydrocarbons retaining the 9 carbons from the original TMB with either even H numbers (closed shell) or odd H numbers (open shell) with limited amount of products from fragmentation, i.e. ions with less than nine C. $C_9$ compounds with an O/C ratio of 6/9 or higher were classified as HOM monomers with the general formula $C_9H_{12-16}O_{6-11}$ in the mass range 280 – 360 $m/z$. Oxygenated hydrocarbons found in the range 460 – 560 $m/z$ containing 18 C were classified as dimers with chemical formulas $C_{18}H_{24-30}O_{10-16}$. The monomer with the highest

intensity detected was $C_9H_{14}O_7$ $m/z$ 296. The highest intensities among the dimers were $C_{18}H_{26}O_{10}$, $C_{18}H_{28}O_{11}$ and $C_{18}H_{28}O_{12}$ at $m/z$ 464, $m/z$ 482 and $m/z$ 498, respectively. In addition to HOM monomers and dimers, nitrogen containing compounds were found as $C_9$ compounds with one or two N or $C_{18}$ compounds with one N. The nitrogen containing compounds were of the general formulas $C_9H_{12-18}NO_{6-13}$, $C_9H_{12-18}N_2O_{8-15}$ and $C_{18}H_{18-24}NO_{6-10}$. The dominating ON were $C_9H_{13}NO_8$ at $m/z$ 325, $C_9H_{15}NO_{10}$ at $m/z$ 359 and $C_9H_{14}N_2O_{10}$ at $m/z$ 372 respectively. In the

experiments where $NO_x$ was added, the formation of ON compounds was increasing with the $NO_x$ concentration. In parallel, the levels of HOM monomers and dimers were reduced with $NO_x$ concentration, where dimers were stronger affected than monomers. Even if the fragmentation products were limited some fragmentation leading to less than 9 carbons could be observed. The most prominent fragments were assigned molecular formulas $C_4H_7NO_7$ at $m/z$ 243, $C_4H_6O_{12}$ at $m/z$ 246, $C_5H_6O_{12}$ $m/z$ 285 and $C_6H_9NO_7$ at $m/z$ 269. Some compounds with C

numbers of 15 and 17 were detected in the range 270 – 560 $m/z$ but their contribution to the total signal was negligible (Figure 2 ).



**Table 1: Experimental conditions for experiments with 30 ppb TMB. Ozone and initial NOx concentration at time 0 are given in ppb and explicitly modelled OH exposure in molecules s cm$^{-3}$. TMB reacted ($\Delta$ TMB) in ppb after a reaction time of 34 s and particle number concentration given in # cm$^{-3}$ after reaching steady state in Go:PAM. RH in all experiments was 38%.**

| # | $[O_3]_0$ | $[NO_x]_0$ | OH exposure | $\Delta$ TMB | $NO_x$ /$\Delta$TMB | Particle number | Contribution of top 10 species (%) |
|---|---|---|---|---|---|---|---|
| 1 | ~19 | 5 | $3.5 \times 10^9$ | 5.4 | 0.9 | - | 28.6 |
| 2 | ~19 | 5 | $7.1 \times 10^9$ | 9.9 | 0.5 | - | 29.8 |
| 3 | ~100 | 3 | $3.8 \times 10^{10}$ | 26 | 0.1 | 60 ± 14 | 38.6 |
| 4 | ~100 | 3 | $2.1 \times 10^{11}$ | 30 | 0.1 | 1610 ± 217 | 35.9 |
| NOx9 | ~9 | 82 | $6.3 \times 10^9$ | 9 | 9.1 | - | 52.4 |
| NOx3$_L$ | ~12 | 38 | $7.9 \times 10^9$ | 11 | 3.5 | - | 42.3 |
| NOx3$_H$ | ~100 | 79 | $3.1 \times 10^{10}$ | 25 | 3.2 | - | 34.2 |
| NOx1 | ~100 | 35 | $9.1 \times 10^{10}$ | 30 | 1.2 | 170 ± 50 | 30.5 |

The relative contribution of the different compound classes to the total assigned signal are shown in Figure 2. It is apparent that HOM monomers and dimers dominate in the experiments with low NO$_x$. The contribution of the monomers to the total oxidation product signal ranges from 20.7- 42.1% and dimers make 6.8 – 43.3 % of the total, depending on experimental conditions. Dimer contributions are highest at high OH exposure (exp 3 and 4 with estimated OH exposure of $3.8 \times 10^{10}$ and $2.1 \times 10^{11}$ molecules s cm$^{-3}$, respectively) and decrease with increasing NO$_x$. Nitrated compounds dominated the spectra with contributions up to ~75% in the experiments with highest amount of NO$_x$. Surprisingly, some nitrated compounds were also found in the experiments without added NO$_x$ which may stem from background NO contamination (~3-5ppb).





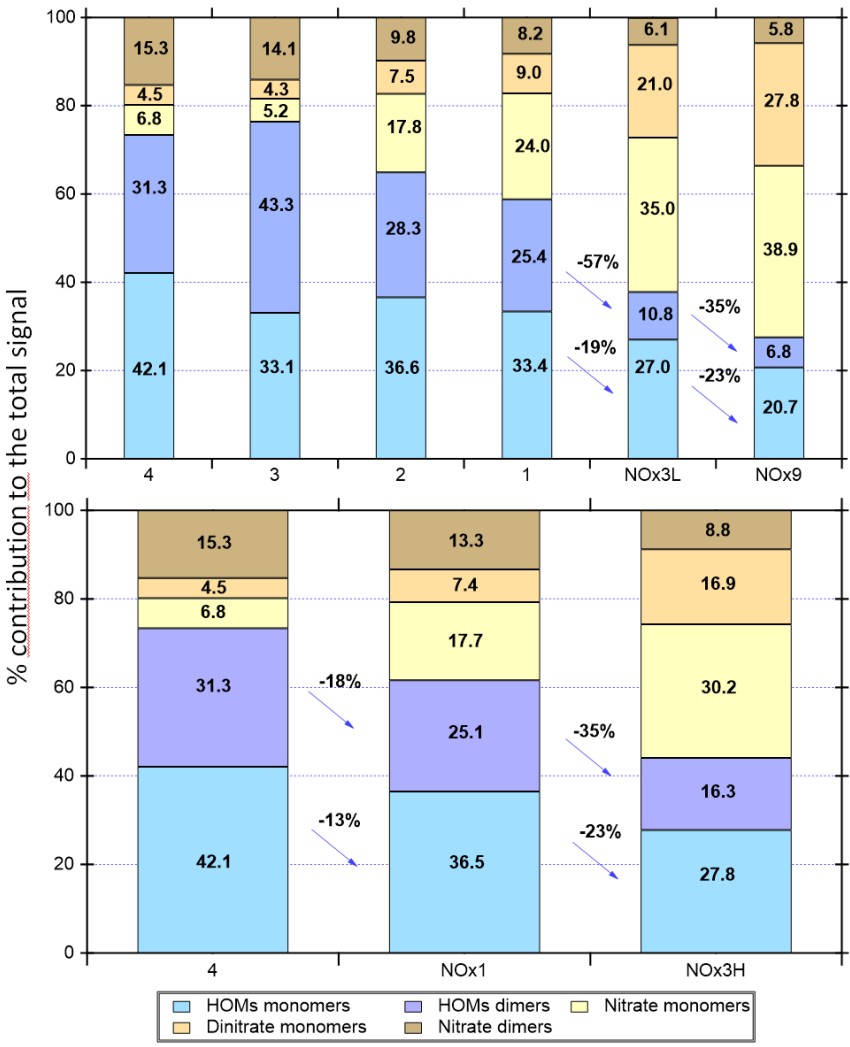

**Figure 2: Overview of different compound groups to the total explained signal. The influence of increased NOₓ/VOC on the product distribution. Top panel illustrates the influence of a decrease of OH exposure (exp 4 - exp 1) and further decrease after adding NOₓ in exp NOx3L – exp NOx9. Dimers show a larger relative reduction than monomers with increasing NOₓ/VOC. Bottom panel shows the influence of increased NOₓ/VOC on the product distribution. Experiments 4, 3 and NOx1 resulted in particle formation.**

Recently, Molteni et al. (2018) assigned 17 compounds making up 80% of the total detected signal for HOM oxidation products from the reaction of TMB with OH. Their compound with the highest fraction of the signal (24.2%) was the dimer $C_{18}H_{26}O_8$. This compound was not detected in our study. We did neither detected deprotonated compounds in the mass range 270 – 560 *m/z* nor HOM monomers with 17 H nor compounds with O/C<0.55 which were found by Molteni et al. (2018). However, we do find 10 of the previously reported 17 monomers and dimers in our spectra. The oxidation product distributions in our experiments are in general term



more diverse, i.e. we found more compounds with smaller yields compared to Molteni et al. (2018). In our experiments the highest 20 compounds together explain 46 – 63 % of the total signal with the individual highest oxidation products contributing only between 4.4 and 16%.

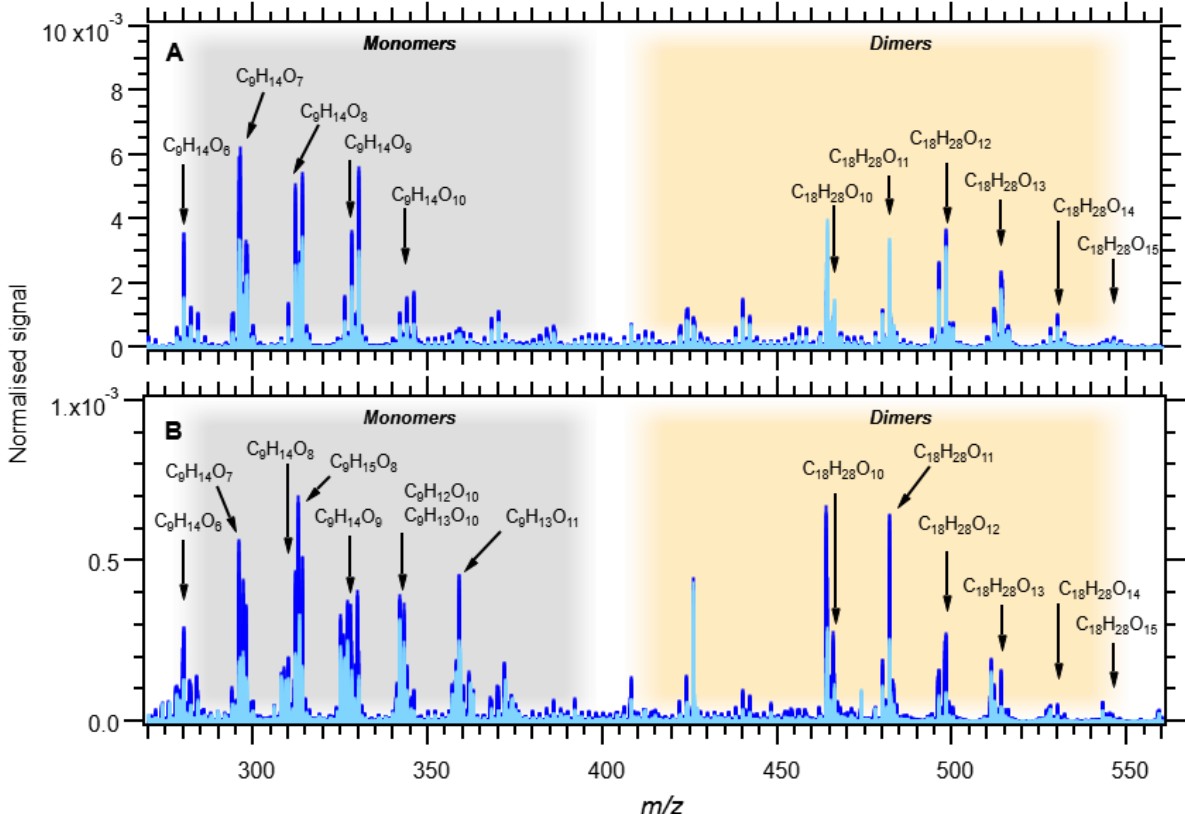

**Figure 3: Mass spectra of all experiments without $NO_x$: 1, 2, 3, and 4. Panel A shows experiments 3 (light blue) and 4 (blue) with OH exposure of $2.87 \times 10^{11}$ and $3.47 \times 10^{11}$ molecules s $cm^{-3}$ respectively. Panel B shows experiment 1 (light blue) and 2 (blue) with of $3.47 \times 10^{10}$ and $7.62 \times 10^{10}$ molecules s $cm^{-3}$ respectively. The signal at m/z 426 is associated with the used mass calibrant PFHA. Note the ten times lower Normalised Signal scale in B.**

The amount of TMB reacted after 34 s in the experiments ranges from 30 ppb (almost all) in experiment 4 to 5 ppb in experiments 1 and NOx9 (comp. Table 1 and Figure S1). The signal intensities of the HOM monomers are increasing with increasing OH exposure. Dimer compounds also increase with increasing OH and reach their highest levels in exp 3, resulting in the highest ratio of dimer to monomer (see Figure 2). Faster conversion of TMB will result in higher initial $RO_2$ levels enabling faster $RO_2 + RO_2$ (self-) reaction. The concentration profile of $RO_2$ in exp 1 and 2 is lower and more evenly spread out over the length of Go:PAM (Figure S1) and the influence of the $RO_2 + RO_2$ reaction will be less in these experiments compared to exp 3 and 4. Enhanced OH exposure does



not only affect the monomer/dimer ratio but also the total amount of compounds measured. Observed compounds were 10 times higher in the high OH exposure exp 3, 4, compared to exp 1, 2. This was also valid for the experiments with added $NO_x$ (NOx1 and $NOx3_H$ *vs* $NOx3_L$ and NOx9). Under high OH exposure (exp 4) the major HOM monomer is $C_9H_{14}O_7$ with a contribution of 5.5 % to the total explained signal, followed by $C_9H_{16}O_9$ and

5 $C_9H_{14}O_8$, contributing 4.6 and 4.5 % respectively.

At the lowest OH exposure (exp 1) the major signals comprise the monomer $C_9H_{12}O_{10}$ and the dimers $C_{18}H_{26}O_{10}$ and $C_{18}H_{28}O_{11}$, contributing 4.4, 4.0 and 3.4 % to the total. These dimer compounds, $C_{18}H_{26}O_{10}$ and $C_{18}H_{28}O_{11}$, are the highest signals in exp 2 and 3. Two open shell species were found among the larger signals in exp 1 and 2 (see Figure 2): $C_9H_{15}O_8$ and $C_9H_{15}O_7$.

The number of H atoms is a characteristic for HOM monomers and dimers. An overview of different oxidation product generations is given in Table 2. We observe HOM monomers with 12-16 H and all compounds with an even number of H are closed shell products. Compounds with an uneven number of H are open shell molecules (radicals). Following the proposed termination scheme by Mentel et al. (2015) compounds with 12 H can be identified as first generation monomers (terminated from $C_9H_{13}O_x$ radicals) and compounds with 16 H as

second generation monomers (terminated from $C_9H_{15}O_x$ radicals). $C_9H_{14}O_x$ can originate from either $C_9H_{13}O_x$ or $C_9H_{15}O_x$. The $RO_2$ radicals $C_9H_{13,15}O_x$ can form dimers likely via the reaction

HOM - $RO_2$ + HOM - $RO_2$ → HOM - dimer + $O_2$                     (3)

Such reaction is possible between any HOM-$RO_2$, first or second generation (and other $C_9$ peroxy radicals with sufficient high abundance). Two first generation HOM-$RO_2$ will result in $C_{18}H_{26}O_x$ dimers while one first and one

second generation HOM-$RO_2$ produce a $C_{18}H_{28}O_x$ and dimerization of two second generation HOM-$RO_2$ will result in dimers of the formula $C_{18}H_{30}O_x$.

     A closer examination of the contribution of different HOM generations in Table 2 shows that first generation monomers with 12 H are showing the highest contribution in experiments with low OH exposure (exp 1, 2) while monomers with 14 H gain importance with higher OH exposure. The second generation monomers with

25 16 H dominate in experiments with the highest OH exposure (exp 3, 4). The dimer population with 28 H has a larger fraction of the total signal at higher OH exposures compared to dimers with 26 H. Dimer population with 30 H is generally lower than other dimers but has the highest fraction in exp 4. The overall dimer fraction of up to 43% in this study (Figure 2) is similar to the dimer fraction of 40% reported by Molteni et al. (2018). However the relative contributions of monomer and dimer generations differ. We find higher contributions of $H_{12}$ monomers (up

to 11%) and a higher contribution of $H_{28}$ dimer (up to 12%) under our experimental conditions. The contribution of $H_{14}$ monomers and $H_{26}$ dimers is significantly less compared to Molteni et al. (2018).

     Increasing OH exposure promotes second OH attacks on oxidation products leading to the observed reduction of first generation products ($C_9H_{12}O_x$) as well as increase of second generation products ($C_9H_{14}O_x$ and $C_9H_{16}O_x$). The increased oxidation degree can also explain the formation of dimers ($C_{18}H_{28}O_x$) from first and second

generation $RO_2$ ($C_9H_{13}O_x$ and $C_9H_{15}O_x$) at higher OH exposure and the increase in second generation dimers ($C_9H_{15}O_x$ + $C_9H_{15}O_x$ → $C_{18}H_{30}O_x$). Open shell species are observed as first generation $RO_2$ ($C_9H_{13}O_x$) and have a higher contribution lower OH exposures. Second generation $RO_2$ ($C_9H_{15}O_x$) have the highest contribution in exp 3. At the highest OH exposures in exp 4, the contribution of ($C_9H_{15}O_x$) is reduced at the expense of $C_9H_{14}O_x$ and $C_9H_{16}O_x$ - HOM and dimers.



**Table 2: Contribution of oxidation product families to the total signal between 270 – 560 *m/z***

| Compound family | 1 | 2 | 3 | 4 | NOx9 | NOx3$_L$ | NOx3$_H$ | NOx1 |
|---|---|---|---|---|---|---|---|---|
| $C_9H_{12}O_x$ | 11.3 | 7.8 | 3.5 | 5.4 | 8.5 | 9.3 | 5.9 | 5.4 |
| $C_9H_{13}O_x$ | 5.7 | 4.5 | 3.3 | 2.1 | 6.0 | 7.0 | 5.6 | 4.2 |
| $C_9H_{14}O_x$ | 8.3 | 10.8 | 9.9 | 17.4 | 4.3 | 6.6 | 8.0 | 13.0 |
| $C_9H_{15}O_x$ | 5.6 | 7.2 | 9.4 | 4.1 | 2.4 | 3.3 | 4.6 | 4.7 |
| $C_9H_{16}O_x$ | 4.8 | 8.0 | 7.7 | 14.5 | 1.5 | 2.5 | 4.8 | 10.3 |
| $C_{18}H_{26}O_x$ | 8.5 | 9.1 | 8.5 | 9.3 | 0.8 | 2.0 | 2.7 | 6.0 |
| $C_{18}H_{28}O_x$ | 7.1 | 9.9 | 9.3 | 11.0 | 0.4 | 1.2 | 2.6 | 6.9 |
| $C_{18}H_{30}O_x$ | 0.7 | 1.0 | 2.3 | 2.6 | 0.4 | 0.4 | 0.9 | 1.7 |
| $C_9H_{13}NO_x$ | 6.2 | 4.7 | 0.6 | 0.6 | 26.8 | 17.1 | 10.1 | 4.8 |
| $C_9H_{15}NO_x$ | 6.4 | 5.6 | 0.7 | 0.9 | 3.7 | 10.3 | 14.5 | 8.5 |
| $C_9H_{14}N_2O_x$ | 2.1 | 1.7 | 1.0 | 1.1 | 18.7 | 11.8 | 7.5 | 1.9 |

In this study the contribution of $C_{18}H_{28}O_x$ shows that both first and second generation HOM-RO$_2$ were dimerising. The kinetics of dimer formation, if produced from RO$_2$ self-reaction, depends on the square of [HOM-RO$_2$ ] and their relative importance will increase with the RO$_2$ concentration. Increased local RO$_2$ concentrations (in the first part of Go:PAM) would explain the increase of dimers with increasing OH exposure.

Substantial particle formation was observed in exp 4, under the highest OH exposure. Although the amount of reacted TMB in exp 3 and 4 is similar (26 and 30 ppb respectively), significant particle formation was not observed under the conditions of exp 3. The rate at which new particle formation (nucleation) occurs is related to the chemical composition and concentration of the nucleating species (McGraw and Zhang, 2008). After reaching the critical nucleus, particle growth becomes spontaneous in the presence of condensable vapour. Apparently, the

local concentration of nucleating species or condensable vapour was not high enough in exp 3 to yield large numbers of particles, compared to exp 4. According to recent studies (Ehn et al., 2014;Trostl et al., 2016;Mohr et al., 2017;McFiggans et al., 2019) dimers play an important role in new particle formation. Mohr et al. (2017) found decreased levels of gas phase dimers in ambient air during NPF events, which is in line with our observations of lower dimer levels in the presence of particles in exp 4, compared to exp 3. A large enough concentration of low

volatility dimers obviously helps forming critical nuclei that then grow by condensation. Note that the newly formed particles will provide an additional sink for dimers and thus reduce their presence as observable gas phase products at the end of the flow reactor.





**Influence of NOx**

In the experiments NOx1, $NOx3_H$, $NOx3_L$ and NOx9, the $NO_x$ levels were increased (Table 1). The presence of $NO_x$ gave nitrogen containing $C_9$ compounds with one or two N atoms and $C_{18}$ compounds with only one N atom, in addition to HOM monomers and HOM dimers. N-containing compounds were the dominating species in
experiments with $NO_x$ (see Figure 2), except for the experiment NOx1 with the smallest amount of $NO_x$ added and a high OH exposure (NOx1). The amount of nitrated compounds increased with the amount of added $NO_x$ at the expense of HOM monomers and HOM dimers as illustrated by the arrows in Figure 2. The effect of the added $NO_x$ is attenuated at higher OH exposure. E.g., the nitrated monomers are reduced from 35.0 and 38.9% at low OH exposure down to 17.7 and 30.2% at high OH exposure. Under elevated $NO_x$ conditions nitrated species were found
among the 10 compounds with the highest contribution to the respective total signal, (see the top-ten lists in SI Table 1 and Figure 4).

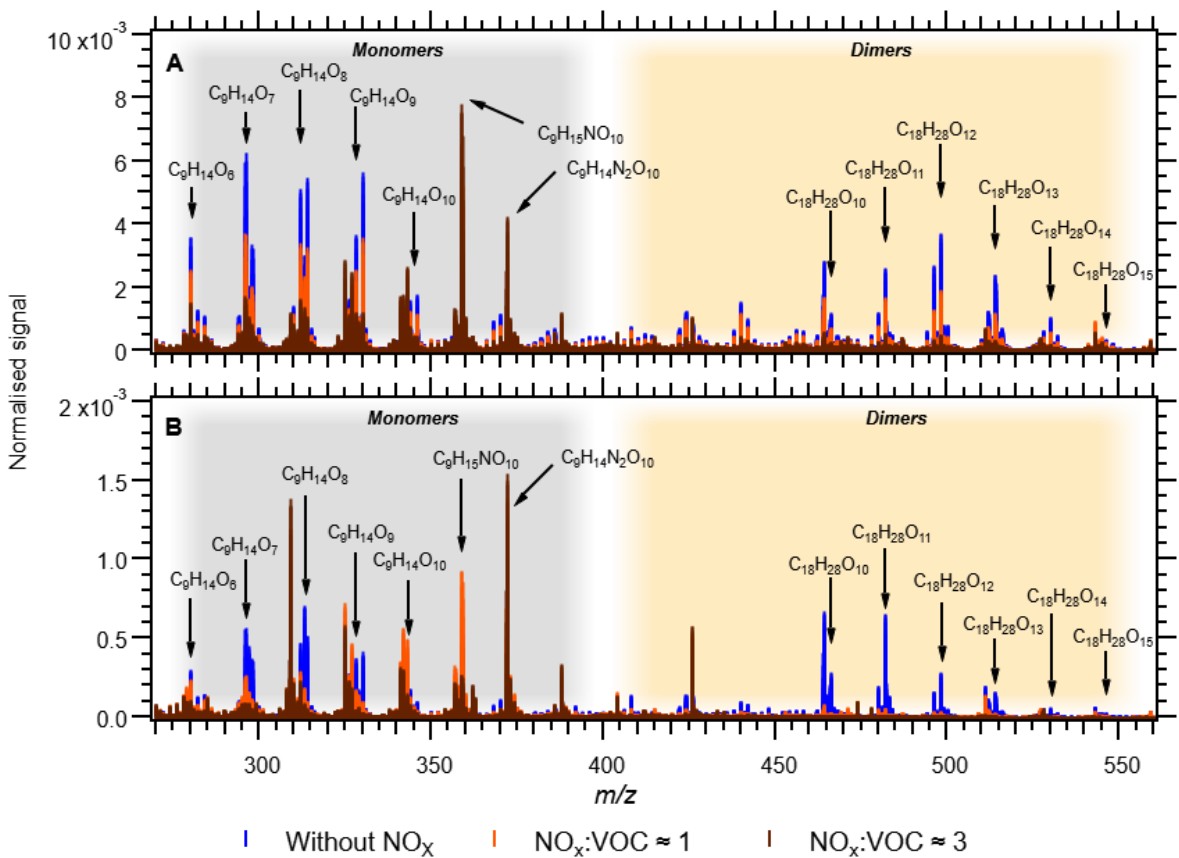

**Figure 4: Comparison of mass spectra of HOM and nitrates.Panel A shows experiment  4 (blue), NOx1**
**(orange) and experiment $NOx3_H$ (brown). Panel B shows experiments 2 (blue), $NOx3_L$ (orange) and NOx9**
**(brown).**



For the two experiments with low OH exposure where the nitrated species dominated, seven out of the top ten species were nitrated with the highest signal arising from the di-nitrated compound ($C_9H_{14}N_2O_{10}$). In these two experiments the top-ten compounds contributed the most to the observed signal (42.3 and 52.4%); most likely owing to the high fraction of nitrated species acting as radical chain termination products. Under highest NOx conditions, the high OH exposure experiment ($NOx3_H$) has six nitrated compounds in the top-ten list with the di-nitrated $C_9H_{14}N_2O_{10}$ on the second rank. In experiment NOx1 only one nitrated species ($C_9H_{15}NO_{10}$) is found in the top-ten list despite the elevated NOx conditions.

Dimer formation is drastically reduced in the presence of elevated $NO_x$. The $NO_x$ effect on NPF was clear and high $NO_x/\Delta TMB$ supresses NPF from TMB oxidation, a trend that was also observed by Wildt et al. (2014) and Lehtipalo et al. (2018) for NPF from monoterpene oxidation. Whenever the products were dominated by ON ($NOx3_L$, NOx9 and $NOx3_H$) particle formation was not observed. Reduced particle formation was observed in NOx1 compared to exp 1 which had a higher contribution of HOM. Owing to the importance of dimers for NPF, as reported by Lehtipalo et al. (2018), we suggest that the reaction of $RO_2 + NO$ resulting in ON is responsible for the observed reduced particle formation, because it competes with the dimer formation from $RO_2 + RO_2$, This reaction is also reducing the contribution of HOM monomers and dimers with increasing $NO_x/\Delta TMB$ and decreasing OH exposure from a total of 61% in exp NOx1 (OH exposure=$9.1\times10^{10}$ molecules s cm$^{-3}$) to 27.5% in NOx9 (OH exposure=$6.3\times10^9$ molecules s cm$^{-3}$). The reaction $RO_2 + NO$ reduce the amount of HOM monomers and dimers in competing with autoxidation and termination by $RO_2+RO_2$ or $RO_2+HO_2$. The yield of ON from $NO+RO_2$ must be high or maybe close to unity based on the measurable increase of ON and the decrease of $NO_x$ in the system (see Figure 7).

The contribution of HOM monomer generations follows the general trend observed in experiments without $NO_x$. The first generation HOM have higher contribution at low OH exposures in $NOx3_L$ and NOx9 and the contribution of second generation HOM is higher the higher the OH exposure is. Similarly, the contribution of $C_9H_{14}O_x$ is increasing with increasing OH exposure. Analogously to the families of HOM, different families of nitrates can be defined. Table 2 gives an overview of the nitrate families and how they contribute to the total signal in the experiments. At lowest OH exposures we find the highest contributions of first generation nitrates ($C_9H_{13}NO_x$) as well as di-nitrates ($C_9H_{14}N_2O_x$). The contribution of second generation nitrates ($C_9H_{15}NO_x$) is increasing with increasing OH exposure and is highest in $NOx3_H$.

For the formation of observed first generation nitrates $C_9H_{13}NO_{6-12}$ ($C_9H_{13}NO_x$ in Table 2) we propose the reaction of a first generation (HOM-)$RO_2$ with NO following the pathway:

$$C_9H_{13}O_{5-11} + NO \rightarrow C_9H_{13}NO_{6-12} \tag{4}$$

The precursor $C_9H_{13}O_5$ is formed after two autoxidation steps and its termination reaction with NO results in $C_9H_{13}NO_6$ which has only minor contribution. $C_9H_{13}NO_{7\&8}$, with higher contribution in exp $NOx3_L$ and NOx9 can be formed from the radicals $C_9H_{13}O_6$ and $C_9H_{13}O_7$ respectively. The even oxygen number in $C_9H_{13}O_6$ indicates that the compound should have undergone a transformation to RO (via reaction with $RO_2$ or NO) and subsequent H-shift and further $O_2$ addition (Vereecken and Peeters, 2010;Mentel et al., 2015).

Second generation nitrates ($C_9H_{15}NO_{8-12}$) can be formed after an additional OH attack (and introduction of an additional H) on a first generation (HOM) monomer, which explains the increase of these compounds with increasing OH exposure. The termination of the $RO_2$ radical chain with NO (reaction 5) will then lead to the





formation of the second generation nitrate. The formation of $RO_2$ precursor species with lower O numbers, i.e. $C_9H_{15}O_{7-8}$ likely stem from compounds terminated earlier in the radical chain process ($C_9H_{14}O_{4-5}$), which do not fall in the typical HOM class (O/C $\geq$ 6/9). The reaction of a first generation nitrate with OH, followed by autoxidation could also possibly produce $C_9H_{15}NO_{8-12}$ by terminating via hydroxyl (reaction 6) or hydroperoxy pathway (reaction 7).

$$C_9H_{15}O_{7-11} + NO \rightarrow C_9H_{15}NO_{8-12} \tag{5}$$
$$C_9H_{14}NO_{8-12} + RO_2 \rightarrow C_9H_{15}NO_{8-12} + R=O \tag{6}$$
$$C_9H_{14}NO_{8-12} + HO_2 \rightarrow C_9H_{15}NO_{8-12} + O_2 \tag{7}$$

For the formation of the most abundant di-nitrates of the formula $C_9H_{14}N_2O_{8-12}$ (reaction 8), OH has to attack a nitrated compound $C_9H_{13}NO_{6-10}$ and the $RO_2$ radical chain has to be terminated with NO:

$$C_9H_{14}NO_{7-11} + NO \rightarrow C_9H_{14}N_2O_{8-12} \tag{8}$$

The precursor species $C_9H_{14}NO_7$ would be in this case formed from the OH attack on the smallest possible nitrate $C_9H_{13}NO_4$ (formed after one autoxidation step and NO termination).

It should be noted that the formation of peroxy nitrates via the reaction $RO_2 + NO_2 \rightarrow RO_2NO_2$ with alkyl and acyl-$RO_2$ (PAN-like) cannot be ruled out as a potential formation mechanisms of nitrates.

The volatility of the produced ON seems to be too high (compared to dimers) to initiate NPF at the present concentrations as seen from the reduction in particle formation potential from exp 4 (HOM dominated, particle formation) to NOx1 (reduction in HOM and increase in ON, reduced particle formation) and NOx3$_H$ (ON dominated, no particle formation)

Figure 5 shows the results for the evolution of HOM monomer, HOM Dimers and organic nitrates (ON) as calculated using the kinetic model described in detailed in SI. The model used has a very simplified scheme for TMB oxidation and subsequent $RO_2$ chemistry. We implemented a few rate coefficients suggested in the literature in order to demonstrate how those compare with our experimental results. Rate coefficients for $RO_2$ termination reactions from MCM in combination with rate coefficients of dimer formation by Berndt et al. (2018) (case 1 and 2 in SI 1) led to an overestimation of dimer compounds. Better representation of the observations were achieved by applying a) the rate coefficients proposed by Zhang et al. (2018) for dimer formation ($2.0 \times 10^{-12}$ cm$^3$molecules$^{-1}$ s$^{-1}$ ) and $1.0 \times 10^{-12}$ cm$^3$ molecules$^{-1}$ s$^{-1}$ for termination and RO formation with branching ratios of 0.4 and 0.6, respectively; b) the rate coefficient from Berndt et al. (2018) for ON formation from HOMRO$_2$ + NO ($1.0 \times 10^{-11}$ cm$^3$molecules$^{-1}$ s$^{-1}$ , branching ratio 0.3 for reaction 56).

In Figure 5, the calculated HOM dimer contribution are the sum of medium (produced from HOM-$RO_2$+$RO_2$), and highly oxidised dimers (produced from HOM-$RO_2$ + HOM-$RO_2$) while the ON includes both organic nitrates (produced from NO+HOM-$RO_2$) and peroxy nitrates (produced from HOM-$RO_2$+$NO_2$). The increased production of monomers calculated for exp 1 and 2 is in agreement with experimental results, where we observed a slightly larger contribution from monomers compared to dimers. Calculated concentrations for dimers are higher in exp 3 and 4, compared to monomers. Secondary chemistry (reaction of OH with the products and possible formation of a second generation of HOM or nitrates) was not taken into consideration in the model. For the experiment with NO$_x$ (Figure 5, right panels) the modeled product distribution follows the general trend of monomer, dimer and nitrate that we observe in the exp NOx3$_L$ and NOx9 with a general higher nitrate production





compared to monomers and dimers. For the $NO_x$ experiments with high initial ozone, the model can reproduce the higher HOM monomer and dimer levels in NOx1 but slightly overestimates the contribution of dimers. Particle formation was observed in NOx1 which might explain the overestimation owing to missing condensation sink in the model. Modelling $NOx3_H$ gives an overestimation of monomers and dimers.

In $NOx3_H$ modelled dimers start forming after ~15 sec. Almost all NO is converted to ON or $NO_2$ at this point and the reaction $HOM-RO_2+NO$ does not produce additional ON and the modelled levels of ON reach a plateau while contribution of HOM dimers can increase. For exp NOx1 and $NOx3_H$ the model is in better agreement with the measurements if only the HOM - dimer formation (from $HOM-RO_2+HOM-RO_2$) is taken into account, i.e. highly oxidized dimers excluding medium oxidized dimer (from $HOM-RO_2 + RO_2$). Generally, the simplified

model was able to support our analysis of the TMB chemistry as described above. Furthermore, it gave some support to the recently suggested mechanisms included in case 3 (e.g. Zhang et al. (2018))

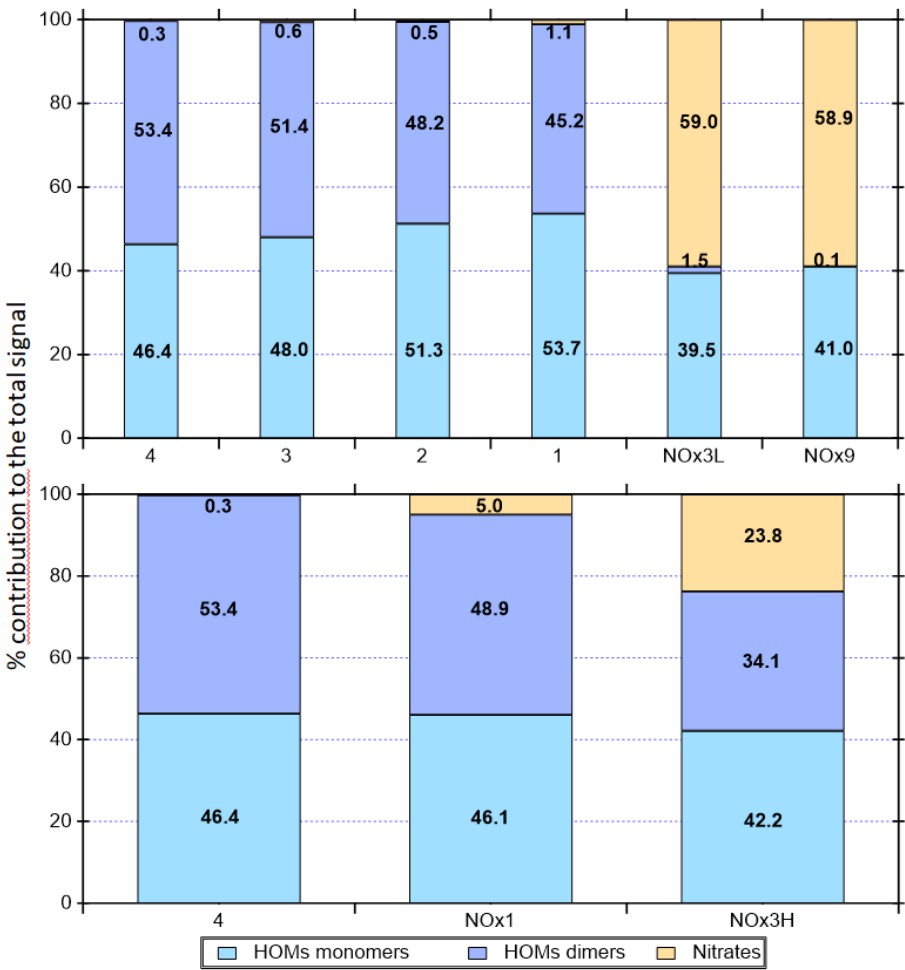

**Figure 5: Modelled product distribution to the total for all 8 experiments (compare Fig. 2).**



**Atmospheric implication and Conclusion**

We have measured the formation of HOM monomers and dimers from OH initiated oxidation of TMB. The experiments with highest OH exposures lead to particle formation when $NO_x$ was not added. With increasing OH exposure and increased likelihood of a second OH attack, we observe a higher contribution from second generation oxidation products and dimers in general. The latter is attributed to the increased $RO_2$ concentrations from the increased/fast TMB consumption by OH. The observed products in this study match what would be expected as termination products from previously proposed reaction mechanisms for HOM formation.

The addition of $NO_x$ to simulate urban condition leads to the formation of ON in addition to HOM and a reduction in particle formation potential. We observe that the formation of ON is increasing with increasing $NO_x/\Delta TMB$, mostly at the expense of dimers. The presence of ONs, formed at the expense of dimers, can explain the decreased tendency for particle formation. We therefore suggest that the reaction HOM-$RO_2$+NO competes with HOM-$RO_2$ self-reaction yielding primarily a reduction in dimer formation, which is responsible for the reduction in particle formation.

According to studies by Molteni et al. (2018) and Wang et al. (2018) HOM formation from AVOCs was observed and consequently AVOC-HOM were suggested as potential contributors to observed NPF in urban atmospheres. In our study, under $NO_x$ free conditions, we found several of previous identified HOM even if we did not fully agree with the identity and relative importance of all HOM. However, the oxidation in polluted environments will happen under elevated $NO_x$ levels and, as has been shown here this can lead to formation of ON instead of HOM and subsequently a reduction in NPF potential. We conclude that for interpretation of NPF from aromatics in urban areas care should be taken and the OH exposure, $NO_x$ levels and $RO_2$ concentrations need to be considered in details since they will largely determine if the HOM-$RO_2$+NO can compete with reactions yielding HOM, and especially HOM dimers.

*Data availability*: The data set is available upon request by contacting Mattias Hallquist (hallq@chem.gu.se).

*Competing Interests*: The authors declare that they have no conflict of interest.

*Author contribution:* Julia Hammes analysed the data and prepared the manuscript with contributions from all co-authors. The experiments were designed by Julia Hammes and Epameinondas Tsiligiannis. Epameinondas Tsiligiannis developed the model code and performed the simulations.

*Acknowledgements*: The research presented is a contribution to the Swedish strategic research area ModElling the Regional and Global Earth system, MERGE. This work was supported by the Swedish Research Council (grant numbers 2014-05332; 2013-06917) and Formas (grant number 2015-1537)



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
