# Peer review of "Effect of NOx on 1,3,5-trimethylbenzene (TMB) oxidation product distribution and particle formation"

_Atmospheric Chemistry and Physics, 2019_

## Referee Comment (RC1) · Anonymous Referee #1 · 6 Jun 2019

General comments:

This manuscript reports results of laboratory experiments on secondary organic aerosol (SOA) formation from the photooxidation of 1,3,5-trimethylbenzene (TMB). TMB is an SOA precursor emitted from anthropogenic sources. The authors employ an original flow reactor combined with a chemical ionization mass spectrometer to investigated effects of OH exposure and NOx level on the distribution of oxidation products including particles, highly oxygenated molecules (HOMs), dimeric HOMs, and nitrated HOMs. They concluded that anthropogenic VOCs such as TMB could lead new particle formation (NPF) but NPF is suppressed under high NO conditions. The research

subject of this study is paid central attention in the field of atmospheric chemistry. The authors employ cutting edge instruments and provide new physical insight into the field of atmospheric chemistry. Because flow reactor experiments under high NOx conditions are very new, the authors should discuss difference between examined reaction conditions and ambient ones. This manuscript suits for the scope of this journal and will be publishable after revisions are made by taking into account reviewer's comments.

Major comments:

(1) Please describe ozone concentration data in the text to discuss the reaction of remaining ozone with NO. If ozone level is higher than 50 ppb, the reaction of ozone with NO (with the rate constant of $1.8 \times 10^{-14}$ cm3 molecule-1 s-1) can significantly occur within a reaction time of 34 s, and NO is converted to NO2. The authors primary assume that nitrated HOMs are formed from the reactions of HOM-RO2 radicals with NO. However, formation of peroxy nitrates from the reactions of HOM-RO2 with NO2 or formation of nitrates from the reactions of HOM-RO with NO2 might be important if NO2 level is much higher than NO level. In experiments with NOx under high ozone levels, NPF was observed. These results may suggest that NO become very low levels due to the reaction with ozone within the reaction time of 34 s, and the reactions of HOM-RO2 with NO2 may become more important than the reactions of HOM-RO with NO.

(2) The authors use NOx levels of 35-82 ppbv, which are similar to NOx levels in urban air, whereas they use HOx (including OH and HO2) levels much higher than ambient levels to accelerate reactions in the laboratory. These conditions may result in overestimation of HOM-RO2 + HO2 reactions and underestimation of HOM-RO2 + NO, HOM-RO2 + NO2, and HOM-RO2 autoxidation, compared to ambient conditions. The authors should discuss difference in branching of HOM-RO2 reactions between present laboratory conditions and ambient conditions.

(3) The authors assume that nitrogen-containing products formed from the oxidation

of TMB in the presence of NOx are nitrates or peroxy nitrates; however this is not evident and further discussion will be necessary in the text. Basically the reviewer agrees with authors' assumption, but in general major nitrogen-containing products, formed from the oxidation of aromatic hydrocarbons, are nitro-aromatic compounds. 1,3,5-Trimethylbenzene is highly methyl-substituted aromatic hydrocarbon and multiple methyl groups inhibit formation of nitro-aromatic compounds (Sato et al., 2012). At this point of view, TMB employed in this study is not a typical aromatic hydrocarbon and a specific molecule, which barely lead to formation of nitro-aromatic compounds.

Specific comments:

(4) Page 3, line 14 and reference list in page 18. Sato et al., 2018 should be Sato et al., 2012

(5) Page 3, line 32. In the unit, "L mol-1", "-1" should be superscript.

(6) Page 4, lines 1-5. Please discuss the phase of products detected by APi-TOF-MS. If it detects products in the aerosol phase, how were these particulate products vaporized in the ion source? Brief explanations would be necessary in the text.

(7) Page 9, lines 14-15. The formation process of compounds with 12 H and 16 H should be explained more in detail. For example, the words, "(terminated from $C9H13Ox$ radicals)", should be written as "(formed from the $C9H13Ox$ + $RO2$ -> $C9H12Ox-1$ + $ROH$ + $O2$ reaction)".

(8) Page 9, lines 32-34. $C9H14Ox$ products include first-generation and second-generation products, i.e., these are formed from the $C9H13Ox+1$ + $RO2$ -> $C9H14Ox$ + R'CHO + $O2$ reaction as well as $C9H15Ox+1$ + $RO2$ -> $C9H14Ox$ + $ROH$ + $O2$ reaction. Please explain why $C9H14Ox$ products have mainly characteristics of second-generation products.

(9) Page 9, last sentence. The authors describe "the contribution of $C9H15Ox$ is reduced at expense of $C9H14Ox$ and $C9H16Ox$ – HOM and dimers," but the meaning

of this sentence is unclear. Do the authors mean that the contribution of C9H15Ox is reduced at the expense of C9H14Ox and C9H16Ox?

(10) Page 12, line 20. The reviewer cannot find "Figure 7."

(11) Page 13, line 4. The description, "hydroxyl", should be "RO2".

(12) Page 13, eq. (6). The chemical formula, R=O, would not be accurate. If the authors use "R=O", please explain that R=O represents carbonyl products.

(13) Page 13, line 29. Please correct "reaction 56)".

(14) Page 15, line 15. "Wang et al. (2018)" should be "Wang et al. (2017)".

(15) The caption of FigureS1. In the caption it is explained "Bottom: Modelled product distribution for all 8 experiments", but the reviewer cannot find this bottom figure.

(16) Table S2. The symbol, "=", represents reversible reaction if it is used in reaction equations. The reviewer recommends for the authors to use arrow symbols instead.

Reference:

K Sato, A Takami, Y Kato, T Seta, Y Fujitani, T Hikida, A Shimono, T Imamura, AMS and LC/MS analyses of SOA from the photooxidation of benzene and 1, 3, 5-trimethylbenzene in the presence of NO x: effects of chemical structure on SOA aging, Atmospheric Chemistry and Physics, 12, 4667-4682 (2012).

---

## Referee Comment (RC2) · Anonymous Referee #2 · 11 Jun 2019

Review of "Effect of NOx on 1,3,5-trimethylbenzene (TMB) oxidation product distribution and particle formation" by Hammes et al.

This manuscript investigates the formation of highly oxygenated molecules (HOMs) from the oxidation of TMB under different conditions (i.e., OH exposure and NOx concentrations). The HOMs are measured by a $NO_3^-$ CIMS and the distributions of HOMs species under different reaction conditions are reported. As there are few studies on the HOMs from aromatics oxidation, these results are worthy to be documented. The major finding is that NOx inhibits the formation of HOMs and enhances the formation of organonitrates, by altering the $RO_2$ chemistry. The experiments are nicely designed and conducted and the manuscript is clearly written. I recommend publication after major revision.

Major Comments

1. In the kinetic model, the oxidized peroxy radicals (HOMRO2) were considered to be formed after 3 autoxidation steps of a general $RO_2$ with a rate constant of 0.1667 $s^{-1}$. This rate is inferred from α-pinene + $O_3$ $RO_2$. However, the isomerization rate of biogenic $RO_2$ is generally not applicable to aromatic $RO_2$, because of the presence of C-C double bonds in aromatics. The isomerization rate of the TMB-OH-OO to bicyclic alkyl radical could be on the order of 1000 $s^{-1}$ [1-2]. For the second isomerization step (i.e., potentially form a tricyclic alkyl radical), the rate is uncertain, but is likely much larger than 0.17 $s^{-1}$ [3]. Even though there are large uncertainties in the $RO_2$ isomerization rates, more appropriate values should be used. A book chapter by Vereecken et al.[3] has a nice summary on this topic.

   In fact, the tuning of the photon flux to match the measured decay of $O_3$ may be related to the poor representation of $RO_2$ chemistry.

2. The authors compare observed HOMs distribution with that in Molteni et al. (2018) and noticed many important discrepancies. The authors must discuss potential causes for the discrepancies.

3. Reaction schematics on the formation mechanism of key HOMs monomer and dimers should be added. This is clearer than describing the mechanism with words.

Minor Comments

1.       How is NOx added into the reactor? I can't find the information in the method section nor schematic figure.

2.       Page 12 Line 18-20. Figure 7 is not in the manuscript. Also, it is highly unlikely that the ON yield from $RO_2$+NO is closed to unity.

3.       Page 12 Line 32. $C_9H_{13}O_5$ is formed after one isomerization step (i.e., initial OH addition, $O_2$ addition, $RO_2$ isomerization, and $O_2$ addition), not two steps.

4.       Page 13 Line 1. What do authors mean by "the formation of RO2 precursor species with lower O numbers"? $C_9H_{15}O_{7-8}$ has many O atoms.

5.       Page 13 Line 26. Zhang et al. (2018) is not in the reference list.

6.       Figure 5 should be combined with figure 2 somehow to facilitate the comparison.

Reference

1.      Wang, S.; Wu, R.; Berndt, T.; Ehn, M.; Wang, L. Formation of Highly Oxidized Radicals and Multifunctional Products from the Atmospheric Oxidation of Alkylbenzenes. *Environ Sci Technol* **2017,** *51*, 8442-8449.
2.      Glowacki, D. R.; Wang, L.; Pilling, M. J. Evidence of Formation of Bicyclic Species in the Early Stages of Atmospheric Benzene Oxidation. *The Journal of Physical Chemistry A* **2009,** *113*, 5385-5396.
3.      Vereecken, L., Reaction Mechanisms for the Atmospheric Oxidation of Monocyclic Aromatic Compounds. In *Advances in Atmospheric Chemistry*, pp 377-527.

---

## Author Response (AR1)

**Authors response on Referee #1-2**

Text in blue is original comment from referee followed by a reply and the actions done to improve the manuscript. A track-changes version of manuscript and supplemental are find at the end of the pdf.

**Response to Anonymous Referee #1**

General comments

This manuscript reports results of laboratory experiments on secondary organic aerosol (SOA) formation from the photooxidation of 1,3,5-trimethylbenzene (TMB). TMB is an SOA precursor emitted from anthropogenic sources. The authors employ an original flow reactor combined with a chemical ionization mass spectrometer to investigated effects of OH exposure and NOx level on the distribution of oxidation products including particles, highly oxygenated molecules (HOMs), dimeric HOMs, and nitrated HOMs. They concluded that anthropogenic VOCs such as TMB could lead new particle formation (NPF) but NPF is suppressed under high NO conditions. The research subject of this study is paid central attention in the field of atmospheric chemistry. The authors employ cutting edge instruments and provide new physical insight into the field of atmospheric chemistry. Because flow reactor experiments under high NOx conditions are very new, the authors should discuss difference between examined reaction conditions and ambient ones. This manuscript suits for the scope of this journal and will be publishable after revisions are made by taking into account reviewer's comments.

**Major comments:**
(1) Please describe ozone concentration data in the text to discuss the reaction of remaining ozone with NO. If ozone level is higher than 50 ppb, the reaction of ozone with NO (with the rate constant of 1.8x10ˆ-14 cm3 molecule-1 s-1) can significantly occur within a reaction time of 34 s, and NO is converted to NO2. The authors primary assume that nitrated HOMs are formed from the reactions of HOM-RO2 radicals with NO. However, formation of peroxy nitrates from the reactions of HOM-RO2 with NO2 or formation of nitrates from the reactions of HOM-RO with NO2 might be important if NO2 level is much higher than NO level. In experiments with NOx under high ozone levels, NPF was observed. These results may suggest that NO become very low levels due to the reaction with ozone within the reaction time of 34 s, and the reactions of HOM-RO2 with NO2 may become more important than the reactions of HOM-RO with NO.

**Reply:** Yes, the titration of NO with ozone is a concern. However, as shown in a new Figure (S3) there is still some NO left at the end of the flow reactor. i.e. after 34 s there is still NO available to react with any HOM-RO$_2$.

In the experiment with NOx in which NPF was observed, the NOx concentration was the lowest and the OH exposure was the highest among the experiments with NOx (Table 1). In addition, the ten compounds with the highest contribution includes only one nitrogen-containing compound, in contrast to the other NOx experiments where the nitrogen-containing compounds were the majority of the top ten (Table S1). This makes us to conclude that the OH concentration was high enough to lead to the formation of more oxidized products, by allowing a second OH reaction to take place, and the formation of dimers which are more

likely to contribute to NPF rather than products from the reaction of HOM-RO$_2$ with NO or NO$_2$.

**Action:** A new Figure (S3) is added to the SI. New text is added to elucidate the effect of titration and the remaining NO.

"As already has been described NO$_x$ was introduced to the Go:PAM as NO. After the addition of ozone, the ozone concentration decreases from 100 ppb to ~80 ppb at the experiment with lower NO$_x$ levels and to ~50 ppb at the experiment with higher NO$_x$ levels, as it reacts with NO producing NO$_2$. For both high and low NO$_x$ conditions there is NO left after the initial reaction with ozone (see grey areas of Figure S3)."

[Figure]

**Figure S3: Particle number (red), ozone (blue), NO$_x$ (black), NO (light blue) and NO$_2$ (cyan) concentrations under NO$_x$ free conditions (top panel), initial NO$_x$:VOC≈1 (middle panel) and NO$_x$:VOC≈3 (bottom panel). The grey areas represent dark conditions, the light yellow, orange and brown represent lower OH exposure conditions (using one UV lamp) and the dark yellow, orange and brown represent higher OH exposure conditions (using two UV lamps).**

(2) The authors use NOx levels of 35-82 ppbv, which are similar to NOx levels in urban air, whereas they use HOx (including OH and HO2) levels much higher than ambient levels to accelerate reactions in the laboratory. These conditions may result in overestimation of HOM-RO2 + HO2 reactions and underestimation of HOM-RO2 + NO, HOM-RO2 + NO2, and HOM-RO2 autoxidation, compared to ambient conditions. The authors should discuss difference in branching of HOM-RO2 reactions between present laboratory conditions and ambient conditions.

**Reply:** Yes, there could be an overrepresentation of HOx chemistry relative to NOx. However, the idea is to indicate a shift from HOx to NOx chemistry so the effect observed might be even more pronounced in polluted environments.

**Action:** New text is added in the atmospheric implication and conclusion section:

"The experimental designed using the Go:PAM with concentrations of HOx (and ROx) higher than ambient would attenuate the influence of added NOx. This will further emphasis the implication of our findings and most likely the NOx effect would be even more important in the urban atmosphere."

(3) The authors assume that nitrogen-containing products formed from the oxidation of TMB in the presence of NOx are nitrates or peroxy nitrates; however this is not evident and further discussion will be necessary in the text. Basically the reviewer agrees with authors' assumption, but in general major nitrogen-containing products, formed from the oxidation of aromatic hydrocarbons, are nitro-aromatic compounds. 1,3,5-Trimethylbenzene is highly methyl-substituted aromatic hydrocarbon and multiple methyl groups inhibit formation of nitro-aromatic compounds (Sato et al., 2012). At this point of view, TMB employed in this study is not a typical aromatic hydrocarbon and a specific molecule, which barely lead to formation of nitro-aromatic compounds.

**Reply:** We agree with the reviewer that nitro-aromatic compounds are quite unlikely to be formed from TMB (Sato et al., 2012). Aromaticity will get lost before initial products react with NO. That's why we assume that's addition of NOx leads to organonitrates ($RONO_2$) or peroxy nitrates ($ROONO_2$), with higher contribution from the $RONO_2$. We also agree that a schematic with proposed reaction mechanisms for the formation of organonitrates will be beneficial to illustrate this.

**Action:** A figure with a proposed formation mechanism of organonitrates is now included (Figure 5).

The following sentences are included:

"These compounds are expected to be nitrates or peroxy nitrates, as it is highly unlikely to form nitro-aromatic compounds from TMB (Sato et al., 2012)."

"A proposed detailed reaction mechanism is depicted in Figure 5."

**Figure 5: The proposed radical reaction mechanism for the formation of some of the mono- and di-nitrates from TMB mentioned in this study. Two separate mechanisms were suggested for the species with the formula $C_9H_{13}NO_8$, which formation pathways are based on (a) (Wang et al., 2017) and (b) (Molteni et al., 2018).**

**Specific comments:**
(4) Page 3, line 14 and reference list in page 18. Sato et al., 2018 should be Sato et al., 2012
**Action:** Done.

(5) Page 3, line 32. In the unit, "L mol-1", "-1" should be superscript.
**Action:** Done.

(6) Page 4, lines 1-5. Please discuss the phase of products detected by APi-TOFMS. If it detects products in the aerosol phase, how were these particulate products vaporized in the ion source? Brief explanations would be necessary in the text.
**Reply:** Only the gas phase oxidation products are detected by the APi-TOFMS.
**Action:** It is now clarified in the text that the detected products are measured in the gas-phase. "Gas phase oxidation products were measured…"

(7) Page 9, lines 14-15. The formation process of compounds with 12 H and 16 H should be explained more in detail. For example, the words, "(terminated from C9H13Ox radicals)", should be written as "(formed from the C9H13Ox + RO2 -> C9H12Ox-1 + ROH + O2 reaction)".
**Reply:** ok

**Action:** Selected reactions have been added to the text. See Reactions 3-15.

$$C_9H_{13}O_x + RO_2 \rightarrow C_9H_{12}O_{x-1} + ROH + O_2 \tag{3}$$

$$C_9H_{15}O_x + RO_2 \rightarrow C_9H_{16}O_{x-1} + R'CHO + O_2 \tag{4}$$

$$C_9H_{13}O_{x+1} + RO_2 \rightarrow C_9H_{14}O_x + R'CHO + O_2 \tag{5}$$

$$C_9H_{15}O_{x+1} + RO_2 \rightarrow C_9H_{14}O_x + ROH + O_2 \tag{6}$$

$$HOM - RO_2 + HOM - RO_2 \rightarrow HOM - DIMER + O_2 \tag{7}$$

$$C_9H_{13}O_x + C_9H_{13}O_x \rightarrow C_{18}H_{26}O_{x-2} + O_2 \tag{8}$$

$$C_9H_{13}O_x + C_9H_{15}O_x \rightarrow C_{18}H_{28}O_{x-2} + O_2 \tag{9}$$

$$C_9H_{15}O_x + C_9H_{15}O_x \rightarrow C_{18}H_{30}O_{x-2} + O_2 \tag{10}$$

$$C_9H_{13}O_{5-11} + NO \rightarrow C_9H_{13}NO_{6-12} \tag{11}$$
$$C_9H_{15}O_{7-11} + NO \rightarrow C_9H_{15}NO_{8-12} \tag{12}$$

$$C_9H_{14}NO_{8-12x} + RO_2 \rightarrow C_9H_{15}NO_{8-12} + Carbonyl\ products \tag{13}$$

$$C_9H_{14}NO_{8-12} + HO_2 \rightarrow C_9H_{15}NO_{8-12} + O_2 \tag{14}$$

$$C_9H_{14}NO_{7-11} + NO \rightarrow C_9H_{14}N_2O_{8-12} \tag{15}$$

(8) Page 9, lines 32-34. C9H14Ox products include first-generation and second generation products, i.e., these are formed from the C9H13Ox+1 + RO2 -> C9H14Ox + R'CHO + O2 reaction as well as C9H15Ox+1 + RO2 -> C9H14Ox + ROH + O2 reaction. Please explain why C9H14Ox products have mainly characteristics of second generation products.

**Reply:** That's true $C_9H_{14}O_x$ products can be either first or second generation products. They have mainly characteristics of secondary products, as their contribution to the top ten compounds (Table S1) increases for the experiments with higher OH exposures, in which more oxidized products can be formed.

**Action:** The text now includes the following:

"$C_9H_{14}O_x$ can be either first or second generation products and originate from either $C_9H_{13}O_x$ or $C_9H_{15}O_x$ (reactions 5 and 6)."

"The $C_9H_{14}O_x$ products have mainly characteristics of second generation products, as their contribution is enhanced in the experiments with higher OH exposures (Table S1), in which there is an enhanced possibility for secondary chemistry initiated by reaction of OH with the first generation products."

(9) Page 9, last sentence. The authors describe "the contribution of C9H15Ox is reduced at expense of C9H14Ox and C9H16Ox – HOM and dimers," but the meaning of this sentence is unclear. Do the authors mean that the contribution of C9H15Ox is reduced at the expense of C9H14Ox and C9H16Ox?

**Reply:** Maybe unclear but we were arguing that the formation of secondary products ($C_9H_{14}O_x$ & $C_9H_{16}O_x$), due to higher OH exposure, is responsible for the reduction of the $C_9H_{15}Ox$ radicals.

**Action:** We modify the text as following: "At the highest OH exposures in exp 4, the contribution of $C_9H_{15}O_x$ radicals, one of the top ten contributors to the signal (Table S1), is reduced, while the contribution of the second generation products ($C_9H_{14}O_x$ and $C_9H_{16}O_x$) and dimers increases."

(10) Page 12, line 20. The reviewer cannot find "Figure 7."
**Reply:** Figure 7 was missed out but should have been found in the supplemental as Figure S3.
**Action:** The missed out Figure 7 is now included in the Supplementary Information as Figure S3.

(11) Page 13, line 4. The description, "hydroxyl", should be "RO2".
**Action:** …Via "peroxy"… is included in the text.

(12) Page 13, eq. (6). The chemical formula, R=O, would not be accurate. If the authors use "R=O", please explain that R=O represents carbonyl products.
**Action:** "R=O" replaced by "Carbonyl Products".

$$C_9H_{14}NO_{8-12x} + RO_2 \rightarrow C_9H_{15}NO_{8-12} + Carbonyl\ products \qquad (13)$$

(13) Page 13, line 29. Please correct "reaction 56)".
**Reply:** The "reaction 56" refers to the corresponding reaction in Table S2. The rate coefficient that is used is based on the recommended one from Berndt et. (2018)
**Action:** The text has been corrected accordingly.
"b) the rate coefficient ($1.0 \times 10^{-11}$ $cm^3 molecules^{-1} s^{-1}$) from Berndt et al. (2018) for HOMRO$_2$ + NO with a branching ratio of 0.3 for ON formation (reaction 56 in Table S2)."

(14) Page 15, line 15. "Wang et al. (2018)" should be "Wang et al. (2017)".
**Action:** Done.

(15) The caption of FigureS1. In the caption it is explained "Bottom: Modelled product distribution for all 8 experiments", but the reviewer cannot find this bottom figure.
**Reply:** The caption "Bottom: Modelled…experiments with NOx." is referring to the Figure 5 in the main text.
**Action:** It is now corrected.
"Figure S1: Vertical profile of TMB (ppb) in the PAM chamber without (left) and with $NO_x$ (right)."

(16) Table S2. The symbol, "=", represents reversible reaction if it is used in reaction equations. The reviewer recommends for the authors to use arrow symbols instead.
**Reply:** We agree.
**Action:** Done.

**Response to Anonymous Referee #2**

General comments

This manuscript investigates the formation of highly oxygenated molecules (HOMs) from the oxidation of TMB under different conditions (i.e., OH exposure and NOx concentrations). The

HOMs are measured by a NO3- CIMS and the distributions of HOMs species under different reaction conditions are reported. As there are few studies on the HOMs from aromatics oxidation, these results are worthy to be documented. The major finding is that NOx inhibits the formation of HOMs and enhances the formation of organonitrates, by altering the RO2 chemistry. The experiments are nicely designed and conducted and the manuscript is clearly written. I recommend publication after major revision.

**Major comments:**
1. In the kinetic model, the oxidized peroxy radicals (HOMRO2) were considered to be formed after 3 autoxidation steps of a general $RO_2$ with a rate constant of 0.1667 $s^{-1}$. This rate is inferred from $\alpha$-pinene + $O_3$ $RO_2$. However, the isomerization rate of biogenic $RO_2$ is generally not applicable to aromatic $RO_2$, because of the presence of C-C double bonds in aromatics. The isomerization rate of the TMB-OH-OO to bicyclic alkyl radical could be on the order of 1000 $s^{-1}$ [1-2]. For the second isomerization step (i.e., potentially form a tricyclic alkyl radical), the rate is uncertain, but is likely much larger than 0.17 $s^{-1}$ [3]. Even though there are large uncertainties in the $RO_2$ isomerization rates, more appropriate values should be used. A book chapter by Vereecken et al.[3] has a nice summary on this topic.
In fact, the tuning of the photon flux to match the measured decay of $O_3$ may be related to the poor representation of $RO_2$ chemistry.
**Reply:** Yes, this part could be more elaborated. A comment is that even if the first step is very fast the following two steps will be rate limiting steps reducing the rate of the overall three step process and might be approaching our rather slow reaction rate used. However, we have now considered this more thoughtfully and the resulting rate of the reaction used in the simplified model has been increased with a factor of 2 (however, the product distributions are more or less in line with original model and support our general conclusions).

The considerations were:
According to the oxygen content in the majority of the $C_9$ products the oxidized peroxy radicals (HOMRO$_2$) should contain either seven, nine or eleven oxygens which would be formed after two, three and four autoxidation steps, respectively. To simplify the model the produced HOMRO$_2$ in the model were assumed to be formed after 3 autoxidation steps. As pointed out by the referee it is not as simple as each step has the same rate coefficient. However, there are large uncertainties where our best estimate would be the following assumptions. The 1st step where the $O_2$ group make a bicyclic radical has a large rate coefficient where Jenkin et al, 2019 suggests a rate coefficient for similar reaction to be larger than 3.6 x $10^2$ $s^{-1}$ (Jenkin et al., 2019). For the 2nd step we assume an internal hydrogen shift potentially facilitated by a conjugated three carbon system. Here Wang et al., 2017 give a large range in reaction rates for similar reactions where the radical from toluene is slow (2.6 x $10^{-2}$) while the radical from larger compounds has higher values (e.g. 7.0 $s^{-1}$ ). We use a value of 1 $s^{-1}$ to represent this step. For the final 3rd step that would represent another hydrogen shift we use the value of 0.5 $s^{-1}$ originally suggested in the paper by Ehn et al., 2014. The combined rate of these three subsequent steps would then be 0.33 $s^{-1}$. This value is a factor of 2 higher than the original value but does not dramatically change our conclusions from the model experiments.
**Action:** The rate of this reaction has been increased in the model and new text has been added to describe the motivation of using such rate for this parametrized three step reaction.
"According to the oxygen content in the majority of the $C_9$ products the oxidized peroxy radicals (HOMRO$_2$) should contain either seven, nine or eleven oxygen which would be formed

after two, three or four autoxidation steps, respectively. To simplify the model the produced HOMRO$_2$ in the model were assumed to be formed after 3 autoxidation steps. There are large uncertainties on estimating the rate coefficients for the autoxidation step (Jenkin et al., 2019). The following assumptions were taken in account for our best lumped estimation of the three step oxidation. The 1$^{st}$ step where the O$_2$ group make a bicyclic radical most likely has a large rate coefficient where Jenkin et al., 2019 suggests a rate coefficient for similar reactions to be larger than 3.6 x 10$^2$ s$^{-1}$ (Jenkin et al., 2019). For the 2$^{nd}$ step we assume an internal hydrogen shift potentially facilitated by a conjugated three carbon system. Here Wang et al., 2017 give a large range in reaction rates for similar reactions where the radical from toluene is slow (2.6 x 10$^{-2}$ s$^{-1}$) while the radical from larger compounds has higher values (e.g. 7.0 s$^{-1}$). We use a value of 1 s$^{-1}$ to represent this 2$^{nd}$ step. For the 3$^{rd}$ step that would represent another hydrogen shift we use the value of 0.5 s$^{-1}$ originally suggested in the paper by Ehn et al., 2014. The combined rate of these three subsequent steps would then be 0.33 s$^{-1}$."

The relative contribution of the HOM monomers and dimers as well as the nitrated compounds using the new rate coefficient is shown in incorporated in the new Figure 2 (Panel B).

"Calculated concentrations for dimers are similar in exp 3 and higher in exp 4, compared to monomers."

[Figure]

**Figure 1: A) Overview of different compound groups to the total explained signal. Top panel illustrates the influence of a decrease of OH exposure (exp 4 - exp 1) and further decrease after adding NO$_x$ in exp NOx3L – exp NOx9. Dimers show a larger relative reduction than monomers with increasing NO$_x$/VOC. Bottom panel shows the influence of increased NO$_x$/VOC on the product distribution. Experiments 4, 3 and NOx1 resulted in particle formation. B) Modelled product distribution shown as lumped categories of nitrated compounds, HOM monomers and dimers and their relative contributions.**

**2. The authors compare observed HOMs distribution with that in Molteni et al. (2018) and noticed many important discrepancies. The authors must discuss potential causes for the discrepancies.**

**Reply:** The experimental conditions and set up are the most important reasons. The Molteni et al. study react only a fraction of the TMB and would thus not form so many secondary products as in the current study. Their initial TMB concentration is 3 times higher (100ppb) compared to our study (30ppb) while the residence time is almost half, 20 sec compared to 34 sec. This may explain the formation of more oxidized compounds, especially more oxidized dimers. In addition, in Molteni et al. the OH radicals are produced outside of the flow tube (see Figure 1 at Molteni et al., 2018), then pre-mixed with the sample flow before the flow reactor. This, in conjunction with the higher TMB initial concentration, may lead to an early consumption of the OH radicals in the initial part of the flow reactor (see Figure S1-1, Supplement at Molteni et al., 2018) minimizing further oxidation, while in our study the OH radicals are produced inside the flow reactor, allowing a higher and more evenly distribution of OH radical concentrations in the flow reactor also favoring secondary reactions.

The formation of new particles in the flow reactor can change the HOM distribution, especially the dimers (Mohr et al., 2017), due to condensational sink. In our study one can see this effect in Figure 3. The product distribution depends on the OH concentration levels as well as on the particle number concentration. A direct comparison to Molteni et al. is not possible, as no values for the particle number concentration are reported. But for the given values of initial TMB (100ppb) and OH (higher than our highest value) concentrations a high particle concentration is expected. This large particle concentration may increase the dimer loss, change the product distribution pattern and make the comparison to our study more difficult. The relative humidity is another parameter which was different during the two studies (75% in Molteni et al., 38% in this study).

Despite these differences the conclusion is the same in both studies, that TMB under NOx free conditions can rapidly form HOM of very low volatility, as they can initiate NPF.

**Action:** The following text has been added to explain the potential discrepancies.

"These differences can be the result of different experimental conditions and set up. In our study the residence time is almost the double compared to Molteni et al., leading to the formation of more oxidized compounds, especially more oxidized dimers, which have been reported in this study. In addition, we produce OH radicals in the full length of the flow reactor enhancing the effects of secondary chemistry. Despite these differences there is a general agreement on the conclusions for NOx free conditions with the Molteni study where one rapidly form HOM of very low volatility, that can initiate NPF."

**3. Reaction schematics on the formation mechanism of key HOMs monomer and dimers should be added. This is clearer than describing the mechanism with words.**

**Action:** Reaction equation have been included (Reactions 3-15), as well as a figure (Figure 5) with a proposed reaction mechanism for the formation of organonitrates.

$$C_9H_{13}O_x + RO_2 \rightarrow C_9H_{12}O_{x-1} + ROH + O_2 \tag{3}$$

$$C_9H_{15}O_x + RO_2 \rightarrow C_9H_{16}O_{x-1} + R'CHO + O_2 \tag{4}$$

$$C_9H_{13}O_{x+1} + RO_2 \rightarrow C_9H_{14}O_x + R'CHO + O_2 \tag{5}$$

$$C_9H_{15}O_{x+1} + RO_2 \rightarrow C_9H_{14}O_x + ROH + O_2 \tag{6}$$

$$HOM - RO_2 + HOM - RO_2 \rightarrow HOM - DIMER + O_2 \tag{7}$$

$$C_9H_{13}O_x + C_9H_{13}O_x \rightarrow C_{18}H_{26}O_{x-2} + O_2 \tag{8}$$

$$C_9H_{13}O_x + C_9H_{15}O_x \rightarrow C_{18}H_{28}O_{x-2} + O_2 \tag{9}$$

$$C_9H_{15}O_x + C_9H_{15}O_x \rightarrow C_{18}H_{30}O_{x-2} + O_2 \tag{10}$$

$$C_9H_{13}O_{5-11} + NO \rightarrow C_9H_{13}NO_{6-12} \tag{11}$$
$$C_9H_{15}O_{7-11} + NO \rightarrow C_9H_{15}NO_{8-12} \tag{12}$$

$$C_9H_{14}NO_{8-12x} + RO_2 \rightarrow C_9H_{15}NO_{8-12} + Carbonyl\ products \tag{13}$$

$$C_9H_{14}NO_{8-12} + HO_2 \rightarrow C_9H_{15}NO_{8-12} + O_2 \tag{14}$$

$$C_9H_{14}NO_{7-11} + NO \rightarrow C_9H_{14}N_2O_{8-12} \tag{15}$$

"A proposed detailed reaction mechanism is depicted in Figure 5."

**Figure 5: The proposed radical reaction mechanism for the formation of some of the mono- and di-nitrates from TMB mentioned in this study. Two separate mechanisms were suggested for the species with the formula C₉H₁₃NO₈, which formation pathways are based on (a) (Wang et al., 2017) and (b) (Molteni et al., 2018).**

**Minor comments**

1. How is NOx added into the reactor? I can't find the information in the method section nor schematic figure.

**Action:** "…while NO was introduced via a NO gas cylinder." is added to the main text.

2. Page 12 Line 18-20. Figure 7 is not in the manuscript. Also, it is highly unlikely that the ON yield from $RO_2+NO$ is closed to unity

**Reply:** Figure 7 was missed out but should have been found in the supplemental as Figure S3.

**Action:** A new Figure S3 is now included in the Supplementary Information. New text reads: "The yield of ON from $NO+RO_2$ might be high based on the measurable increase of ON and the decrease of $NO_x$ in the system (see **Error! Reference source not found.**)."

[Figure]

**Figure S3: Particle number (red), ozone (blue), $NO_x$ (black), NO (light blue) and $NO_2$ (cyan) concentrations under $NO_x$ free conditions (top panel), initial $NO_x$:VOC≈1 (middle panel) and $NO_x$:VOC≈3 (bottom panel). The grey areas represent dark conditions, the light yellow, orange and brown represent lower OH exposure conditions (using one UV lamp) and the**

**dark yellow, orange and brown represent higher OH exposure conditions (using two UV lamps).**

3. Page 12 Line 32. $C_9H_{13}O_5$ is formed after one isomerization step (i.e., initial OH addition, $O_2$ addition, $RO_2$ isomerization, and $O_2$ addition), not two steps.
**Reply:** That's true.
**Action:** New text reads: "one autoxidation step"

4. Page 13 Line 1. What do authors mean by "the formation of RO2 precursor species with lower O numbers"? $C_9H_{15}O_{7-8}$ has many O atoms.
**Reply:** Yes, this is a relative term referring to comparison with other systems, yielding higher number of oxygen but was not clear from the content.
**Action:** We have now modified the text.
"The formation of $RO_2$ precursor species with 7-8 O numbers, i.e. $C_9H_{15}O_{7-8}$ likely stem from compounds terminated earlier in the radical chain process ($C_9H_{14}O_{4-5}$), which do not fall in the typical HOM class (O:C≥ 6:9)."

5. Page 13 Line 26. Zhang et al. (2018) is not in the reference list.
**Action:** Done. It is actually Zhao et al. (2018).

6. Figure 5 should be combined with figure 2 somehow to facilitate the comparison.
**Reply:** Ok.
**Action:** Figure 5 is now merged with Figure 2.

[Figure]

**Figure 2: A) Overview of different compound groups to the total explained signal. Top panel illustrates the influence of a decrease of OH exposure (exp 4 - exp 1) and further decrease after adding NOₓ in exp NOx3L – exp NOx9. Dimers show a larger relative reduction than monomers with increasing NOₓ/VOC. Bottom panel shows the influence of increased NOₓ/VOC on the product distribution. Experiments 4, 3 and NOx1 resulted in particle formation. B) Modelled product distribution shown as lumped categories of nitrated compounds, HOM monomers and dimers and their relative contributions.**

[revised manuscript text omitted]

*Case 1*

| | | | | |
|---|---|---|---|---|
| 63 | HOMRO2 + RO2 →= 0.4 (MONOMER + Carbonyl/Alcohol + O2) | $8.8\times10^{-13}$ | MCM | |
| 64 | HOMRO2 + RO2 →= 0.6 (HOMRO + RO + O2) | $8.8\times10^{-13}$ | MCM | |
| 65 | HOMRO2 + RO2 →= MODIMER + O2 | $8.0\times10^{-11}$ | Medium Oxidized dimer, Berndt et al. (2018) | |
| 66 | HOMRO2 + HOMRO2 →= 0.4 (MONOMER + MONOMER + O2) | $8.8\times10^{-13}$ | MCM | |
| 67 | HOMRO2 + HOMRO2 →= 0.6 (HOMRO + HOMRO + O2) | $8.8\times10^{-13}$ | MCM | |
| 68 | HOMRO2 + HOMRO2 →= HODIMER + O2 | $2.6\times10^{-10}$ | Highly Oxidized dimer, Berndt et al. (2018) | |

*Case 2*

| | | | | |
|---|---|---|---|---|
| 63 | HOMRO2 + RO2 →= 0.4 (MONOMER + Carbonyl/Alcohol + O2) | $1.0\times10^{-12}$ | Zhao et al. (2018) | |
| 64 | HOMRO2 + RO2 →= 0.6 (HOMRO + RO + O2) | $1.0\times10^{-12}$ | Zhao et al. (2018) | |
| 65 | HOMRO2 + RO2 →= MODIMER + O2 | $8.0\times10^{-11}$ | Medium Oxidized dimer, Berndt et al. (2018) | |
| 66 | HOMRO2 + HOMRO2 →= 0.4 (MONOMER + MONOMER + O2) | $1.0\times10^{-12}$ | Zhao et al. (2018) | |
| 67 | HOMRO2 + HOMRO2 →= 0.6 (HOMRO + HOMRO + O2) | $1.0\times10^{-12}$ | Zhao et al. (2018) | |

| | 68 | HOMRO2 + HOMRO2 $\rightarrow=$ HODIMER + O2 | $2.6\times10^{-10}$ | Highly Oxidized dimer, Berndt et al. (2018) |
| Case 3 | | | | |
| | 63 | HOMRO2 + RO2 $\rightarrow=$ 0.4 (MONOMER + Carbonyl/Alcohol + O2) | $1.0\times10^{-12}$ | Zhao et al. (2018) |
| | 64 | HOMRO2 + RO2 $\rightarrow=$ 0.6 (HOMRO + RO + O2) | $1.0\times10^{-12}$ | Zhao et al. (2018) |
| | 65 | HOMRO2 + RO2 $\rightarrow=$ MODIMER + O2 | $2.0\times10^{-12}$ | Medium Oxidized dimer, Zhao et al. (2018) |
| | 66 | HOMRO2 + HOMRO2 $\rightarrow=$ 0.4 (MONOMER + MONOMER + O2) | $1.0\times10^{-12}$ | Zhao et al. (2018) |
| | 67 | HOMRO2 + HOMRO2 $\rightarrow=$ 0.6 (HOMRO + HOMRO + O2) | $1.0\times10^{-12}$ | Zhao et al. (2018) |
| | 68 | HOMRO2 + HOMRO2 $\rightarrow=$ HODIMER + O2 | $2.0\times10^{-12}$ | Highly Oxidized dimer, Zhao et al. (2018) |

**References (supplemental)**

Atkinson, R., Baulch, D. L., Cox, R. A., Hampson, R. F., Kerr, J. A., and Troe, J.: Evaluated Kinetic and Photochemical Data for Atmospheric Chemistry Supplement-Iv - Iupac Subcommittee on Gas Kinetic Data Evaluation for Atmospheric Chemistry, J Phys Chem Ref Data, 21, 1125-1568, Doi 10.1063/1.555918, 1992.

Berndt, T., Scholz, W., Mentler, B., Fischer, L., Herrmann, H., Kulmala, M., and Hansel, A.: Accretion Product Formation from Self- and Cross-Reactions of RO2 Radicals in the Atmosphere, Angewandte Chemie (International ed. in English), 57, 3820-3824, 10.1002/anie.201710989, 2018.

Ehn, M., Thornton, J. A., Kleist, E., Sipila, M., Junninen, H., Pullinen, I., Springer, M., Rubach, F., Tillmann, R., Lee, B., Lopez-Hilfiker, F., Andres, S., Acir, I. H., Rissanen, M., Jokinen, T., Schobesberger, S., Kangasluoma, J., Kontkanen, J., Nieminen, T., Kurten, T., Nielsen, L. B., Jorgensen, S., Kjaergaard, H. G., Canagaratna, M., Maso, M. D., Berndt, T., Petaja, T., Wahner, A., Kerminen, V. M., Kulmala, M., Worsnop, D. R., Wildt, J., and Mentel, T. F.: A large source of low-volatility secondary organic aerosol, Nature, 506, 476-479, 10.1038/nature13032, 2014.

FACSIMILE for Windows 4, v. 4.0.48; MCPA Software Ltd: Faringdon, UK:-, 2009.

Finlayson-Pitts, B. and- Pitts., J.: Chemistry of the Upper and Lower Atmosphere, Academic Press, 1999.

Jenkin, M. E.; Saunders, S. M.; Wagner, V.; Pilling, M. J., Protocol for the development of the Master Chemical Mechanism, MCM v3 (Part B): tropospheric degradation of aromatic volatile organic compounds. Atmos. Chem. Phys. -2003, 3, 181-193, 2003.

Jenkin, M. E.; Valorso, R.; Aumont, B.; Rickard, A.R., Estimation of rate coefficients and branching ratios for reactions of organic peroxy radicals for use in automated mechanism construction. Atmos. Chem. Phys., 19, 7671-7717, 2019.

Li, R.; Palm, B. B.; Ortega, A. M.; Hlywiak, J.; Hu, W.; Peng, Z.; Day, D. A.; Knote, C.; Brune, W. H.; de Gouw, J. A.; Jimenez, J. L., Modeling the Radical Chemistry in an Oxidation Flow Reactor: Radical Formation and Recycling, Sensitivities, and the OH Exposure Estimation Equation. The Journal of Physical Chemistry A -2015, 119, (19), 4418-4432, 2015.

Sander, S. P.; Friedl, R. R.; Barker, J. R.; Golden, D. M.; Kurylo, M. J.; Wine, P. H.; Abbat, J. P. D.; Burkholder, J. P.; Kolb, C. E.; Moortgat, G. K.; Huie, R. E.; Orkin, V. L., Chemical Kinetics and Photochemical data for Use in Atmospheric Studies. JPL publication 10-6 2011, 17 (10-6), 2011.

Wang, S., Wu, R., Berndt, T., Ehn, M., and Wang, L.: Formation of Highly Oxidized Radicals and Multifunctional Products from the Atmospheric Oxidation of Alkylbenzenes, Environ Sci Technol, 51, 8442-8449, 10.1021/acs.est.7b02374, 2017.

Watne, A. K., Psichoudaki, M., Ljungstrom, E., Le Breton, M., Hallquist, M., Jerksjo, M., Fallgren, H., Jutterstrom, S., and Hallquist, A. M.: Fresh and Oxidized Emissions from In-Use Transit Buses Running on Diesel, Biodiesel, and CNG, Environmental science & technology, 52, 7720-7728, 10.1021/acs.est.8b01394, 2018.

Zhao, Y., Thornton, J. A., and Pye, H. O. T.: Quantitative constraints on autoxidation and dimer formation from direct probing of monoterpene-derived peroxy radical chemistry, Proc Natl Acad Sci U S A, 115, 12142-12147, 10.1073/pnas.1812147115, 2018.

---

## Author Response (AR2)

**Authors response on minor comments frpm Referee #2**

Text in blue is original comment from referee followed by a reply and the actions done to improve the manuscript. A track-changes version of manuscript is find at the end of the pdf.

(1) Although this is not a comment for the manuscript, we cannot see a reference cited in response to minor comments 2 of referee #2.

**Reply:** This was a field code linking to a Figure. In parenthesis it was written: "(see Figure S3)" but after moving Figures there was an error message. Thus, it is not a reference. In the manucsripth there was no error.

(2) Lines 84-85. The authors wrote, "while favouring ON formation (reaction 1)", but it may still lead readers' misunderstanding. Can you suggest here that the yield of reaction 1 is 0.3 at most?

**Action:** Bold text was added to the sentence: " "while favouring **some** ON formation (reaction 1 **with a yield of up to 0.3**)"

(3) Line 175. The m/z of C5H6O12 should be 258.

**Action:** Done.

(4) Lines 210-216. The authors irradiated 254 nm light, which might induce the photolysis of aldehydes and peroxides, to the full length of the flow reactor; this is also a difference from Molteni et al.

**Reply:** We do not agree on adding a speculation on photolysis of aldehydes/peroxides. The estimated photon flux ($1.6 \times 10^{16}$ photons $cm^{-2} s^{-1}$) combined with typical aldehyde/peroxide cross sections of $10^{-20}$ $cm^2$ $molecule^{-1}$ will give lifetimes of several thousands of seconds (the residence time is 30s). However, we can add a statement on that the OH was produced by irradiation at 254 nm.

**Action:** "In addition we produce OH radicals **through irradiation at 254 nm** in the full length of the flow reactor enhancing the effects of secondary chemistry**.**" (bold text was added)

(5) Line 278. The word, "increases", might be "increased".

**Action:** Done.

(6) Line 334. The expression, "the yield of ON from NO+RO2 might be high", is again vague. Can you suggest here that the yield of reaction 1 is 0.3 at most?

**Action:** "the yield of ON from NO+RO2 might be **significant (e.g. up to 0.3)**" (bold text was changes)

(7) Line 373. Please revise equation 13. The reactant, "C9H14NO8-12x", should be "C9H14NO9-13". The term, "+ O2", should be added to the left hand side.

**Action:** Done (8) Line 411. The words, ", right panels", could be removed.

**Action:** Done (9) Lines 440-443. From present results, can the authors suggest a region of NO level, in which the found transition occurs in urban air?

**Reply:** Unfortunately, this cannot be directly derived from our experiments. The region would be when the NO reaction is competing with autoxidation and for sure that will be at rather low NOX but to really pin-point this one need to do a kinetic study.

**Action:** No action

[revised manuscript text omitted]